# Reproducibility and implementation of a rapid, community-based COVID-19 "test and respond" model in low-income, majority-Latino communities in Northern California

**Gabriel Chamie** [1]*, **Patric Prado**[1], **Yolanda Oviedo**[2], **Tatiana Vizcaíno**[3], **Carina Arechiga**[1], **Kara Marson** [1], **Omar Carrera**[2], **Manuel J. Alvarado**[3], **Claudia G. Corchado**[3], **Monica Gomez**[3], **Marilyn Mochel**[3], **Irene de Leon**[2], **Kesia K. Garibay**[4], **Arturo Durazo**[4], **Maria-Elena De Trinidad Young**[4], **Irene H. Yen**[4], **John Sauceda** [1], **Susana Rojas**[5], **Joe DeRisi**[1], **Maya Petersen** [6], **Diane V. Havlir**[1], **Carina Marquez**[1]

1 University of California, San Francisco, San Francisco, California, United States of America, 2 Canal Alliance, San Rafael, California, United States of America, 3 United Way-Merced, Merced, California, United States of America, 4 University of California, Merced, Merced, California, United States of America, 5 Latino Task Force for COVID-19, San Francisco, California, United States of America, 6 University of California, Berkeley, Berkeley, California, United States of America

* Gabriel.Chamie@ucsf.edu

**Data Availability Statement:** Data for this study are publicly available at Harvard Dataverse (https://dataverse.harvard.edu): Prado, Patric, 2022,

## Abstract

### Objective

To evaluate implementation of a community-engaged approach to scale up COVID-19 mass testing in low-income, majority-Latino communities.

### Methods

In January 2021, we formed a community-academic "Latino COVID-19 Collaborative" with residents, leaders, and community-based organizations (CBOs) from majority-Latinx, low-income communities in three California counties (Marin/Merced/San Francisco). The collaborative met monthly to discuss barriers/facilitators for COVID-19 testing, and plan mass testing events informed by San Francisco's Unidos en Salud "test and respond" model, offering community-based COVID-19 testing and post-test support in two US-census tracts: Canal (Marin) and Planada (Merced). We evaluated implementation using the RE-AIM framework. To further assess testing barriers, we surveyed a random sample of residents who did not attend the events.

### Results

Fifty-five residents and CBO staff participated in the Latino collaborative. Leading facilitators identified to increase testing were extended hours of community-based testing and financial support during isolation. In March-April 2021, 1,217 people attended mass-testing events over 13 days: COVID-19 positivity was 3% and 1% in Canal and Planada, respectively. The RE-AIM evaluation found: census tract testing coverage of 4.2% and 6.3%, respectively;

"Replication Data for: Title: Reproducibility and implementation of a rapid, community-based COVID-19 "test and respond" model in low-income, majority-Latino communities in Northern California", https://doi.org/10.7910/DVN/K4BQCl.

**Funding:** GC and CM received a grant from the National Institutes of Health (https://www.nih.gov/) Acceleration of Diagnostics – Underserved Populations (RADx-UP): Award Number P30AI027763. The funders had no role in the study design, data collection and analysis, decision to public, or preparation of the manuscript.

**Competing interests:** The authors have declared that no competing interests exist.

90% of event attendees were Latino, 89% had household income <\$50,000/year, and 44% first-time testers (reach), effectiveness in diagnosing symptomatic cases early (median isolation time: 7 days) and asymptomatic COVID-19 (41% at diagnosis), high adoption by CBOs in both counties, implementation of rapid testing (median: 17.5 minutes) and disclosure, and post-event maintenance of community-based testing. Among 265 non-attendees surveyed, 114 (43%) reported they were aware of the event: reasons for non-attendance among the 114 were insufficient time (32%), inability to leave work (24%), and perceptions that testing was unnecessary post-vaccination (24%) or when asymptomatic (25%).

## Conclusion

Community-engaged mass "test and respond" events offer a reproducible approach to rapidly increase COVID-19 testing access in low-income, Latinx communities.

## Introduction

Timely and accessible COVID-19 testing is critical to limiting COVID-19 transmission and ensuring access to effective therapies. With prompt diagnosis, persons with COVID-19 can isolate, limiting transmission [1], and meet eligibility periods for effective therapies to prevent hospitalization and death [2]. Community-based testing sites that are easily accessible and that offer high-quality testing are essential to meeting COVID-19 testing needs.

From the start of the COVID-19 pandemic, inequities have contributed to disparities in COVID-19 testing access among historically underserved, low-income, Latino communities in the United States, compared to predominantly non-Latino White and high-income communities [3]. These disparities have persisted despite the disproportionate burden of COVID-19 infections and morbidity faced by Latino communities [4]. In California, Latino persons make up 38.9% of the state's population but accounted for 47.8% of reported COVID-19 cases and 44.2% of reported COVID-19 deaths by March 2022 [5]. Reproducible and rapidly scalable strategies to address COVID-19 testing disparities through flexible, community-partnered approaches are needed, particularly in the context of variable testing demand and intermittent surges caused by SARS-CoV-2 variants, such as delta and omicron.

Early in the COVID-19 pandemic in April 2020, a community-academic partnership in San Francisco, "*Unidos en Salud*/United in Health", rapidly developed and implemented low-barrier, community-based, "test and respond" mass testing events in one of the city's predominantly Latino neighborhoods [6]. Key components of these events included walk-up (i.e., no appointment), rapid throughput (i.e., aiming to minimize time from arrival to completion of site activities and maximize persons seen per day), outdoor testing, and capacity to reach a high proportion of community residents in days. Event planning, mobilization and implementation were delivered in partnership with community-based organizations, organized within San Francisco's Latino Task Force for COVID-19, and adopted key features in response to community input, including multi-lingual staff and information, messaging that testing did not constitute a "public charge," offering testing outside of weekday work hours, and post-test support for persons in isolation or quarantine. *Unidos en Salud*'s initial mass testing events subsequently evolved into standing, drop-in rapid COVID-19 antigen testing and vaccination services at a well-known public transit hub in the Mission District [7, 8].

Informed by San Francisco's *Unidos en Salud* model, we sought to evaluate implementation and reproducibility of applying a community-engaged approach to scaling up rapid, low-

barrier COVID-19 mass "test and respond" events to suburban and rural, low-income, majority-Latino communities in Northern California.

## Methods

### Latino COVID-19 Collaborative

In January 2021, we formed a community-academic "Latino COVID-19 Collaborative" (LCC) with residents and community-based organizations (CBOs) from majority-Latino, low-income communities in two California counties (i.e., Marin, Merced). The objective of the LCC was to foster cross-community collaboration, address shared barriers to COVID-19 testing implementation, and facilitate the rapid development of local COVID-19 "test and respond" programs. We conducted a brief survey among community residents and CBO members at baseline regarding perceived barriers and facilitators to accessing COVID-19 testing in each community. We chose diverse sites–one suburban and one rural–to test reproducibility and implementation of the *Unidos en Salud* approach in settings distinct from urban San Francisco.

In Marin County, north of San Francisco, the Canal district is a predominantly Latino, low-income community with a high proportion of residents who are essential frontline workers and who have recently immigrated to the US, and with densely populated housing [9]. In Merced County–a rural, agricultural county in California's Central Valley–Planada is a low-income, majority-Latino, rural unincorporated community [10] with a mix of year-round residents and seasonal residents (i.e., migratory agricultural workers), predominantly from Latin America. Members of San Francisco's Latino Task Force, a collective of CBOs addressing COVID-19 who held leadership roles in *Unidos en Salud*, also joined the LCC. Academic partners in the collaborative included researchers from UCSF and UC Merced.

The LCC met monthly to discuss barriers and facilitators to COVID-19 testing and to plan mass testing events offering low-barrier, community-based COVID-19 rapid testing with post-test support in US-census tracts in the Canal (tract 1122.01) and Planada (tract 19.01).

### Mass COVID-19 "test and respond" events

Following formation of the LCC, CBO and research staff members of the LCC designed mass testing events with LCC input. Specifically, the events were designed to take place at well-known, central locations (a community park in Canal, and the local middle school and community center in Planada), during days and times that optimized access for essential workers (e.g., weekends, evenings), with bilingual (Spanish-English) staff. Key features included free, rapid throughput, walk-up testing, with no identification required. Prior to the events, LCC members conducted community mobilization activities to highlight these key features and promote COVID-19 testing. These activities included posters, pamphlets and radio announcements, with photos of local residents participating in testing.

During the events, attendees completed a brief pre-testing survey to collect contact information for results disclosure and to ascertain demographics, vaccination status and barriers/facilitators to COVID-19 testing. The testing event laboratory staff conducted real-time, COVID-19 rapid antigen testing (Abbott BinaxNOW™) via outdoor anterior nasal swab collection with confirmatory SARS-CoV-2 PCR testing of asymptomatic, rapid antigen positive attendees: PCR was conducted on a second nasal swab, collected from all attendees. All rapid test results were confirmed by a laboratory supervisor and entered real-time into a web-based application (Primary.Health, San Francisco), via WiFi hotspots, that disclosed results to attendees via text or email and reported all tests to the California Department of Public Health.

Staff contacted all persons testing positive for COVID-19 directly by phone to provide counseling and offer local support services during isolation and quarantine.

## Post-event survey of non-attendees

To further assess testing barriers and awareness and perception of the mass testing events, we surveyed adult residents who did not attend the events. Over eight weeks, starting within two months of the mass testing events, we contacted a random sample of census tract residents via phone (from CBO community lists) and in-person outreach (at high transit sites, such as public parks and markets) until we reached 100–200 adults per community who reported event non-attendance. Staff invited the event non-attendees to participate in a brief survey and COVID-19 testing at a local CBO-run test site established after the events, with the same testing procedures used in the events.

## Analysis

We evaluated implementation outcomes using a Reach, Effectiveness, Adoption, Implementation and Maintenance (RE-AIM) evaluation framework [11]. Metrics for reach included census tract coverage (using geolocation of home addresses in ArcGIS) and representativeness of event attendees (using 2020 US Census Data with 2019 American Community Survey estimates for sex, income and age comparisons), engagement of residents testing for COVID-19 for the first time, awareness of the event among the sample of event non-attendees, and geographic range of event attendees. To evaluate factors associated with no prior COVID-19 testing, we performed multivariate logistic regression with a dependent outcome of no prior testing, adjusting for factors of interest *a priori*, including sex, age, household density, income, essential worker status, ability to isolate/quarantine without job loss, reporting having a primary care provider, testing site and COVID-19 vaccination status. Metrics of effectiveness included time to diagnosis and disclosure of COVID-19, days of isolation post-testing, diagnosis of persons with asymptomatic infection, referral to COVID-19 vaccination and, among persons with COVID-19, offer of supportive services. We evaluated adoption by measuring monthly LCC attendance prior to the events and describing the level of CBO and community volunteer engagement in the events and representativeness of event staff relative to each community. Measures of implementation included fidelity to standard operating procedures, study protocol deviations, and local adaptations made by community partners. Finally, we measured maintenance of easily accessible rapid COVID-19 test and response services at community-based sites post-event.

## Ethics statement

All participants provided verbal informed consent prior to participation. The University of California, San Francisco (UCSF) Committee on Human Research approved the study protocol and served as the institutional review board of record for investigators from the University of California, Merced.

## Results

### Latino COVID-19 Collaborative (LCC)

In December 2020, we recruited and enrolled 55 community residents and CBO staff from Marin and Merced Counties, as well as two San Francisco Latino Task Force members, for participation in the LCC: 23 (42%) were CBO staff, and 32 (58%) were community residents. The LCC met monthly and discussed facilitators and barriers to COVID-19 testing access, delivery,

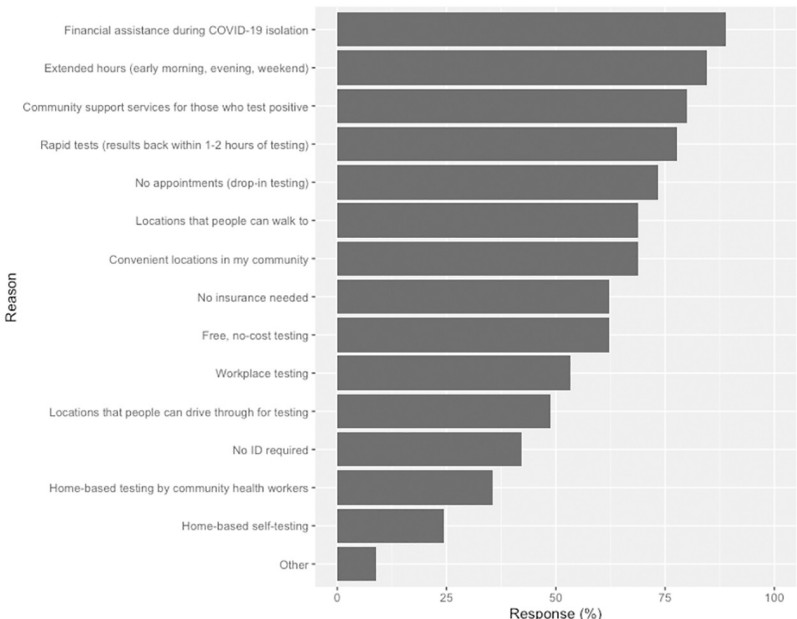

**Fig 1. Facilitators to COVID-19 testing identified by participants in the tri-county Latino COVID-19 Collaborative (N = 45 participants responding to baseline survey) in January 2021.**

and uptake (January 2021), vaccine hesitancy and misinformation (February 2021), and declining COVID-19 testing demand post-vaccine availability (March 2021). The most common facilitators reported in the baseline survey by LCC members to increase COVID-19 testing uptake are shown in Fig 1. The top two facilitators were extended hours for testing (including early morning, late evening, and weekend) and financial support during isolation.

## Mass testing events

Over 13 days in two communities, 1,217 participants enrolled and tested for COVID-19 in the mass test and respond events. In the Canal, 756 persons tested over nine days in March 2021. In Planada, 460 persons tested over four days in April 2021. Among all event attendees, 90% identified as Latino, 63% as essential workers (i.e., required to go to work when stay at home orders were in place), 97% spoke a language other than English at home, and 68% reported household income <$25,000/year (Table 1). Leading testing barriers reported by adult attendees (N = 853 responses to this question) were having no time to get tested (20%), not knowing where to get tested (13%), unsure how to access testing appointments (5%), and losing income after testing positive (4%).

## Post-testing event sampling of non-attendees

From May-July 2021, study staff enrolled 265 (138 Canal and 127 Planada) residents who did not attend the testing events for surveys. Of the 265 non-attendees, 165 tested for COVID-19 at the time of the survey. Among the sample of 265 non-attendees, 97% identified as Latino and 40% as women; median age was 36 years (interquartile range [IQR]: 23–47).

## RE-AIM evaluation

**Reach.** *Census tract coverage & representativeness.* In the Canal and Planada census tracts, 554 and 310 residents attended the mass testing events, reaching 4.2% and 6.3% of US census

**Table 1. Characteristics of mass COVID-19 "test and respond" event attendees from March-April 2021in two predominantly Latino, low-income communities in Northern California.**

| | Canal (N = 756) | Planada (N = 461) | Overall (N = 1,217) |
|---|---|---|---|
| **Age**, median [range] | 32.0 [2.0, 77.0] | 38.0 [2.0, 88.0] | 34.0 [2.0, 88.0] |
| **Sex** | | | |
| Female | 400 (52.9%) | 276 (59.9%) | 676 (55.5%) |
| Male | 342 (45.2%) | 183 (39.7%) | 525 (43.1%) |
| Other | 14 (1.85%) | 2 (0.4%) | 16 (1.3%) |
| **Race** | | | |
| Latino | 673 (89.0%) | 419 (90.9%) | 1092 (89.7%) |
| Asian | 17 (2.3%) | 0 (0%) | 17 (1.4%) |
| Black | 5 (0.7%) | 1 (0.2%) | 6 (0.5%) |
| Non-Latino White | 40 (5.3%) | 13 (2.8%) | 53 (4.4%) |
| Other | 8 (1.0%) | 8 (1.7%) | 16 (1.3%) |
| Unknown | 13 (1.7%) | 20 (4.3%) | 33 (2.7%) |
| **Spanish is the primary language spoken at home** | 373 (94.4%) | 283 (99.6%) | 656 (96.6%) |
| **Essential worker** | 171 (55.9%) | 121 (78.1%) | 292 (63.3%) |
| **Household characteristics** | | | |
| Number of rooms, mean (SD) | 2.17 (1.0) | 3.06 (0.9) | 2.53 (1.1) |
| Number persons per household, mean (SD) | 4.5 (1.6) | 4.0 (1.7) | 4.3 (1.6) |
| **Annual income** | | | |
| <$15,000 | 161 (45.0%) | 52 (21.2%) | 213 (35.3%) |
| $15–20,000 | 67 (18.7%) | 50 (20.4%) | 117 (19.4%) |
| $20–25,000 | 44 (12.3%) | 37 (15.1%) | 81 (13.4%) |
| $25–35,000 | 36 (10.1%) | 31 (12.7%) | 67 (11.1%) |
| $35–50,000 | 20 (5.6%) | 39 (15.9%) | 59 (9.8%) |
| $50–75,000 | 12 (3.4%) | 20 (8.2%) | 32 (5.3%) |
| $75–100,000 | 4 (1.1%) | 8 (3.3%) | 12 (2.0%) |
| >100k | 14 (3.9%) | 8 (3.3%) | 22 (3.7%) |
| **Has a primary medical provider** | 363 (48.1%) | 216 (47.7%) | 579 (48.0%) |
| **Isolation for COVID-19 would result in job loss** | 74 (23.4%) | 22 (14.0%) | 96 (20.3%) |
| **Likert scale questions in response to statements** | | | |
| **"It is easy to get tested for COVID-19."** | | | |
| Strongly agree | 37 (6.9%) | 38 (10.3%) | 75 (8.3%) |
| Agree | 215 (40.3%) | 165 (44.8%) | 380 (42.2%) |
| Neither agree nor disagree | 76 (14.3%) | 35 (9.5%) | 111 (12.3%) |
| Disagree | 53 (9.9%) | 22 (6.0%) | 75 (8.3%) |
| Strongly disagree | 142 (26.6%) | 80 (21.7%) | 222 (24.6%) |
| Declined to answer | 10 (1.9%) | 28 (7.6%) | 38 (4.2%) |
| **"I know where I can get COVID-19 testing in my community."** | | | |
| Strongly agree | 44 (8.2%) | 45 (12.0%) | 89 (9.75%) |
| Agree | 229 (42.6%) | 161 (42.9%) | 390 (42.7%) |
| Neither agree nor disagree | 33 (6.1%) | 23 (6.1%) | 56 (6.1%) |
| Disagree | 66 (12.3%) | 17 (4.5%) | 83 (9.1%) |
| Strongly disagree | 151 (28.1%) | 110 (29.3%) | 261 (28.6%) |
| Declined to answer | 15 (2.8%) | 19 (5.1%) | 34 (3.7%) |

SD, standard deviation.

**Table 2. A comparison of mass testing event attendee and U.S. Census tract demographic characteristics in the two study communities.**

| | Canal | | Planada | |
|---|---|---|---|---|
| | Testing event attendees (census tract residents only) | U.S. Census 2020 | Testing event attendees (census tract residents only) | U.S. Census 2020 |
| N | 341 | 8,024 | 335 | 5,279 |
| Age, years (median)* | 32 | 26 | 38 | 31 |
| Female sex* | 55% | 43% | 58% | 47% |
| Latino ethnicity | 94% | 92% | 92% | 87% |
| Income <$50,000/ year* | 89% | 51% | 97% | 63% |

*Data not available in 2020 U.S. Census: used 2019 American Community Survey

tract residents, respectively. When comparing residents who attended the testing events to 2020 US census tract data, we found that in both communities, attendees were older, lower income and more likely to identify as female (Table 2). The geographic range of event attendees included persons traveling up to 40 miles in the Canal and 68 miles in Planada (Table 3).

**Table 3. RE-AIM implementation evaluation framework of the mass testing event intervention March-April 2021 in two predominantly Latino, low-income communities in Northern California.**

| RE-AIM dimension | Implementation measures | Implementation outcomes |
|---|---|---|
| Reach | 1. Census tract coverage | Canal (U.S. Census Tract 1122.01): 4.2% |
| | | Planada (U.S. Census Tract 19.01): 6.3% |
| | 2. Event attendee representativeness from census tract residents | See Table 2. |
| | 3. Geographic range of attendees (driving distance from reported home address) | Canal: 40 miles |
| | | Planada: 68 miles |
| | 4. Attendees with no prior COVID-19 testing | 534/1,217 (44%) |
| | 5. Awareness of event among a random sample of non-attendees | 114/265 (43%) |
| Effectiveness | 1. COVID-19 positivity | 28/1,217 (2%) |
| | | (Canal 3% and Planada 1%) |
| | 2. % of COVID-19 cases with asymptomatic infection | 11/27 (41%) |
| | 3. Time to delivery of results to persons with COVID-19 | Median (IQR): 71 minutes (15 minutes-29 hours) |
| | 4. Isolation time post-disclosure among persons with symptomatic COVID-19 | Median (IQR): 7 days (5–8 days) |
| Adoption | 1. Latino COVID-19 Collaborative: average attendance at pre-event planning meetings | Mean attendance: 51/55 (93%) |
| | 2. Local community-based organization (CBO)-led planning and coordination of testing events? | Yes, at both sites. |
| | 3. Participation of local community health workers (*promotores*) on testing days? | Yes, at both sites. |
| | 4. Post-test support services provided by local CBOs or county for persons with COVID-19? | Yes, at both sites. |
| Implementation | 1. Implementation according to study protocol | Yes, at both sites, with no protocol deviations. |
| | 2. Mean time from arrival at event to completion of testing | 17.5 minutes (Canal: 22 minutes and Planada: 10 minutes) |
| | 3. Proportion of persons with COVID-19 reached for testing result disclosure | 100% at both sites |
| Maintenance | 1. Continuation of community-based, drop-in testing post-event | Community-based rapid COVID-19 testing delivered by CBOs continued in both communities throughout 2021 post-event |

Among persons who attended the mass testing event, 534 (44%) reported that they had never been previously tested for COVID-19. Correlates of never having been previously tested included greater household density (i.e., persons/number of rooms per household; OR: 1.3, 95% CI: 1.04–1.75, p = 0.02), higher age (OR: 1.02 per year, 95%CI: 1.00–1.04, p = 0.03), not having received a COVID-19 vaccination (OR: 2.56, 95%CI: 1.54–4.32, p<0.001) and testing in rural Planada (vs the Canal, OR: 1.93, 95%CI: 1.11–3.34, p = 0.02).

*Awareness among event non-attendees*. Among the 265 event non-attendees, 114 (43%) reported that they were aware of the mass testing events. Among those aware of the events, the leading reasons provided for mass testing event non-attendance were insufficient time (32%), inability to leave work (24%), and perceptions that testing was unnecessary post-vaccination (24%) or when asymptomatic (25%). Among all non-attendees, nine (3%) reported they would not feel comfortable testing at a public event. Among 165 non-attendees who tested for COVID-19 at the time of completing the post-event survey, 59 (36%) reported that they had never been previously tested for COVID-19.

**Effectiveness.** *Timely diagnosis*. At the two community testing events, 28 (2%) persons tested positive for COVID-19 by BinaxNOW™ rapid antigen testing, with 3% and 1% positivity in the Canal and Planada, respectively. Average positivity over the same period at the county level was 1% in Marin County [12] and 4.6% in Merced County [13]. Among 28 participants with positive rapid antigen test results, 12 were asymptomatic: 10 had positive confirmatory PCR test results, whereas one participant was considered to have had a false-positive rapid test result and another a false-negative PCR. The participant considered to have a false-positive rapid test result was a child in a household with no known COVID-19 infections, who remained asymptomatic for 10-days post-testing, and had a negative confirmatory PCR result. The participant considered to have a false-negative PCR was an adult who tested positive on both repeat rapid antigen testing and repeat PCR confirmatory testing one day after the initial negative PCR result.

Among the 27 persons with confirmed COVID-19, 11 (41%) were asymptomatic at the time of diagnosis. Of 16 persons with symptomatic COVID-19, median time from symptom onset to testing positive was 3 days. Median time to verbal disclosure of results to persons who tested COVID-19 positive by rapid antigen test was 71 minutes (IQR: 15 minutes– 29 hours), and median isolation time post-disclosure among persons with symptomatic COVID-19 was 7 days (IQR: 5–8 days).

**Adoption.** CBO staff in both communities led the Latino COVID-19 collaborative, facilitating all monthly meetings in 2021, with average attendance of 51/55 (93%) at the monthly meetings pre-event. CBO staff also participated mass testing events implementation, as follows.

*The canal*. The overall coordination of the event was led by a local CBO Latina staff member, with support from the Latino chief executive officer. Local, Spanish-speaking community health volunteers (*promotores*) from the Canal mobilized residents on event days. The promotores welcomed participants, answered questions, and assisted with pre-testing surveys. A booth at the event exit offered information about local CBO services, in Spanish and English, to all participants upon completion of testing. Bilingual (Spanish/English) CBO staff, working with the Marin Public Health Department, attended the testing events to ensure all persons testing positive for COVID-19 were reached by phone for disclosure and provided isolation and quarantine guidance and referrals to support services. These staff members also scheduled Marin County vaccination site appointments: 625 community members were provided COVID-19 vaccination appointments at the event.

*Planada*. The coordination of the event was led by a local CBO Latina staff member with expertise in Spanish-English interpretation, with support from the Latino chief executive

officer. Latino *promotores* from Merced County welcomed participants to the events, answered questions and assisted with pre-testing surveys and event navigation. Multiple local CBOs set up booths at the event exit, offering information and materials about services. Unlike the Canal, in which our project hired medical workers to conduct nasal swabs for rapid COVID-19 antigen testing and PCR, in Planada, the school superintendent (a well-known local community leader) and staff from the local middle school who had been previously trained to don/doff personal protective equipment and conduct rapid antigen testing through a California Department of Public Health program, volunteered their time to perform testing at the four-day event. Local CBO staff led disclosure of positive test results, and provision of isolation and quarantine guidance and post-test support, in Spanish and English.

**Implementation.** The two mass testing events occurred as scheduled and were implemented according to the study protocol. No protocol deviations occurred. Average time from registration at the testing site to completion of testing was 22 minutes in the Canal, and 10 minutes in Planada. Staff reached all persons testing positive for COVID-19 for disclosure and offer of post-test support.

Local staff made several adaptations to the events in response to lower than anticipated demand for testing at the time of the mass testing events. In the Canal, staff scheduled vaccine appointments (in high demand at that time [March 2021]) at the testing events. At both sites, local CBO staff instituted an event-based raffle for non-cash prizes (including gift cards to local stores and on-site prizes, such as a television), to promote the testing events.

**Maintenance.** Following testing events and post-event sampling in both communities, low-barrier (i.e., walk-up with no appointments or identification required), community-based rapid COVID-19 testing sites offering rapid antigen testing with PCR confirmation, as well as post-test support services, were maintained by local CBOs (Canal Alliance and United Way-Merced) weekly throughout 2021 and remain ongoing at the time of publication. In the Canal, the community-based testing site has also served as a local COVID-19 vaccination site, with walk-up vaccination or vaccine appointments, on COVID-19 testing days. More recently, facilitated linkage of persons with COVID-19 to outpatient therapy at local clinics has also been implemented from the community-based testing sites in both the Canal and Planada. The LCC continues to meet monthly as well, at the time of publication, supporting community mobilization efforts and communication as the COVID-19 pandemic and public health response continues to evolve.

## Discussion

In this study, we demonstrate, using a RE-AIM implementation evaluation framework, that community-engaged mass "test and respond" events are a reproducible and scalable approach that can be implemented and adapted across a range of settings to rapidly improve access to COVID-19 testing in low-income, Latino communities in Northern California. Key components of this approach, adapted from San Francisco's *Unidos en Salud* model [6, 7], included pre-event collaborative meetings among CBOs, community residents and academic partners, low-barrier, easily accessible testing in well-known locations in each community with response services for persons in isolation and quarantine [14], and transition to sustained community-based testing post-event. Evaluating testing in US census tracts allowed for clear measures of reach, including testing coverage and representativeness of event attendees compared to community residents. Our findings provide insights and evidence for rapidly scaling community-based testing in diverse settings, including rural and suburban sites, with distinct needs, thereby adding to our understanding of strategies to address COVID-19 testing disparities.

In preparation for COVID-19 testing on a large scale, our approach leveraged collaboration across three counties to promote sharing of ideas and to speed dissemination of innovative

solutions to increase testing access and uptake. This community-to-community collaborative model holds the promise to increase COVID-19 testing access and rapid adoption and adaptation of testing procedures in other low-income, majority-Latino communities where limited access persists. In addition, the inclusion of community members and academic partners with experience from the *Unidos en Salud* model in San Francisco, allowed for rapid scale-up of testing based on established and validated procedures [6, 7, 15]. Such cross-community collaborations hold the potential for scale up of other health practices, including improving access to outpatient COVID-19 treatment, as well as other public health priorities in underserved communities.

Our test and respond approach aimed to address distinct structural barriers to COVID-19 testing and related services that have been well-documented among Latino communities in the US [16, 17]. These barriers include low-wage, frontline jobs that do not allow for remote work and limit time for accessing testing and healthcare. In addition, language and internet-access barriers to healthcare services and public health messaging may further limit access to COVID-19 services [18]. Persons who are undocumented may also fear accessing services, and awareness of available services may be lower among recent immigrants [19]. We aimed to systematically address these barriers. We ensured testing event days included weekends and evening hours to optimize access for frontline workers. Likely because of these efforts, most persons who tested (63%) at the events were essential, frontline workers–a group disproportionately affected by the COVID-19 pandemic [6, 17]. We provided mobilization messaging and pre- and post-testing information in Spanish and English, in-partnership with community members and local CBOs with longstanding ties to each community, to address fears and lack of trust. We provided walk-up, no appointment testing, and on-site local volunteers with hand-held, web-connected tablets at testing registration to overcome internet access and navigation barriers. Finally, our post-test response included access to food, cleaning supplies and PPE, to address challenges with loss of income and high-density living environments during isolation and quarantine.

Although our mass testing events took place approximately one year into the COVID-19 pandemic in the US, we identified a high proportion of community members (43%) who had never tested for COVID-19, highlighting challenges in access and uptake to testing prior to our events. The finding that certain factors were associated with increased odds of no prior testing suggests additional nuance to testing barriers within these communities. For example, the association of no prior testing with rural residence may reflect challenges in physically accessing test sites, as has been previously described [20], whereas the association of no prior testing with being unvaccinated may reflect challenges in navigating or trusting the healthcare system [21]. Overall, the high proportion of event attendees with no prior testing and the greater proportion of event attendees who were lower income compared to the underlying census tract populations, suggests that low-barrier, community-based testing approaches hold opportunities to reach persons who have been missed by traditional, health facility-based testing.

Our study used the RE-AIM framework to evaluate a broad range of implementation outcomes [22]. The RE-AIM evaluation yielded several insights. First, by using US census tracts and sampling event non-attendees, we were able to evaluate testing "reach" in multiple ways, including testing coverage and representativeness of event attendees compared to the underlying community population, and gain insights into persistent barriers and penetration of mobilization messaging among persons who opted not to attend the testing events. Second, our measures of effectiveness, including the high median recommended days of isolation post-diagnosis in symptomatic cases (7 days) and prompt disclosure times, suggest potential pathways by which rapid, low-barrier testing can reduce onward transmission [15]. Third, our

implementation measures of time from registration to testing suggest that such testing events can address the commonly reported barrier to testing of having "no time" to test, with rapid (10–22 minute) throughput. Indeed, the high proportion of low-income residents and front-line workers suggests that such events may have addressed this barrier for this at-risk population. The relatively longer throughput (by 12-minutes) in the Canal vs Planada may have been attributable to the Canal being the first event site (with event processes still being learned by the project team) as well as relatively more families testing together in the Canal and self-registering (i.e., completing registration surveys on their own phones for multiple family members, thereby delaying time to completion of testing), though we are unable to verify this as an explanation for the differences in throughput time by site. Lastly, our measures of adoption and maintenance indicate that low-barrier, community-based COVID-19 testing and support services can be rapidly scaled up and continued in diverse low-income, majority-Latino settings when implemented in partnership with local CBOs and community members that reflect the communities offered testing. In addition, the use of local, non-medical workers to conduct testing in Planada was a key adaptation that has since been expanded to the Canal site and may further support rapid testing expansion in times of increased testing and longer-term maintenance. As testing demand has varied considerably over time and effective outpatient treatments become increasingly available, understanding how to rapidly scale, adopt and maintain high-quality, low-barrier testing approaches in such settings is critical to addressing persistent inequities in the COVID-19 pandemic. Such approaches also hold the promise of expansion to address other high-priority prevention and screening services, via multi-disease service offering, as the *Unidos en Salud* project has demonstrated in San Francisco [23], and merit further investigation.

Our study has several limitations. First, we did not measure costs or evaluate cost-effectiveness of the mass testing events. The costs associated with such testing events include costs related to community mobilization efforts, personnel time, and laboratory testing costs, as well as government stimulus funds to assist persons testing positive for COVID-19 that may not continue as the pandemic progresses. However, several studies have found that expanding access to COVID-19 testing at a population level is cost-effective in the US, and can contribute to reductions in infection, hospitalization, and death [24, 25]. Second, we relied on US census tract data to estimate testing reach. Census data may miss certain sub-populations, including undocumented residents, which may have resulted in over-estimation of testing coverage in the two study communities [26]. However, the use of a clear population "denominator" allowed us to gain insights into community coverage and representativeness among persons tested with our testing strategy. Third, our measures of maintenance of testing services are limited to the past year, however testing remains ongoing at the time of publication. Lastly, during our mass testing campaigns, we did not offer PCR testing for persons with negative antigen tests who had symptoms concerning for COVID-19 or close contacts with persons who had COVID-19, and as such we may have missed some persons with early infection with antigen-negative, PCR-positive COVID-19. However, we have previously demonstrated that rapid antigen testing has a high sensitivity for identifying persons with low cycle thresholds on PCR, and in our maintenance phase of testing [7], we have since instituted PCR testing for persons with negative antigen tests who have symptoms concerning for COVID-19 or high-risk exposures.

In summary, community-engaged mass "test and respond" events offer a reproducible approach to rapidly increase COVID-19 testing access in low-income, Latino communities. Our findings provide novel insights into how cross-community collaboration, supported by a community-academic partnership, can address disparities in COVID-19 testing.

## Acknowledgments

We gratefully acknowledge the study participants from the Canal and Planada communities, including the members of the Latino COVID-19 Collaborative. We also wish to thank Marin Health and Human Services in Marin County, California, and the Department of Public Health, Merced County, California.

## Author Contributions

**Conceptualization:** Gabriel Chamie, Omar Carrera, Manuel J. Alvarado, Irene H. Yen, John Sauceda, Maya Petersen, Diane V. Havlir, Carina Marquez.

**Data curation:** Patric Prado.

**Formal analysis:** Gabriel Chamie, Patric Prado, Carina Marquez.

**Funding acquisition:** Gabriel Chamie, Carina Marquez.

**Investigation:** Gabriel Chamie, Omar Carrera, Manuel J. Alvarado, Kesia K. Garibay, Arturo Durazo, Maria-Elena De Trinidad Young, Irene H. Yen, John Sauceda, Joe DeRisi, Maya Petersen, Diane V. Havlir, Carina Marquez.

**Methodology:** Gabriel Chamie, Omar Carrera, Manuel J. Alvarado, Arturo Durazo, Maria-Elena De Trinidad Young, John Sauceda, Susana Rojas, Joe DeRisi, Maya Petersen, Carina Marquez.

**Project administration:** Yolanda Oviedo, Tatiana Vizcaíno, Carina Arechiga, Kara Marson, Omar Carrera, Manuel J. Alvarado, Claudia G. Corchado, Monica Gomez, Marilyn Mochel, Irene de Leon, Irene H. Yen.

**Resources:** Patric Prado, Yolanda Oviedo, Tatiana Vizcaíno, Carina Arechiga, Kara Marson, Monica Gomez, Irene de Leon, Susana Rojas.

**Software:** Patric Prado.

**Supervision:** Gabriel Chamie, Yolanda Oviedo, Tatiana Vizcaíno, Kara Marson, Omar Carrera, Manuel J. Alvarado, Maria-Elena De Trinidad Young, Irene H. Yen, Carina Marquez.

**Validation:** Patric Prado.

**Writing – original draft:** Gabriel Chamie.

**Writing – review & editing:** Gabriel Chamie, Patric Prado, Yolanda Oviedo, Tatiana Vizcaíno, Carina Arechiga, Kara Marson, Omar Carrera, Manuel J. Alvarado, Claudia G. Corchado, Monica Gomez, Marilyn Mochel, Kesia K. Garibay, Arturo Durazo, Maria-Elena De Trinidad Young, Irene H. Yen, John Sauceda, Susana Rojas, Joe DeRisi, Maya Petersen, Diane V. Havlir, Carina Marquez.

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
