## [Decision Letter · Decision Letter 0]

6 Jul 2022

PONE-D-22-13602Reproducibility and Implementation of a Rapid, Community-Based COVID-19 “Test and Respond” Model in Low-Income, Majority-Latino Communities in Northern CaliforniaPLOS ONE

Dear Dr. Chamie,

Thank you for submitting your manuscript to PLOS ONE. After careful consideration, we feel that it has merit but does not fully meet PLOS ONE’s publication criteria as it currently stands. Therefore, we invite you to submit a revised version of the manuscript that addresses the points raised during the review process as far as relevant and feasible.

We look forward to receiving your revised manuscript.

Kind regards,

Benedikt Ley, PhD

Academic Editor

PLOS ONE

Journal Requirements:

Reviewers' comments:

Reviewer's Responses to Questions

**Comments to the Author**

1. Is the manuscript technically sound, and do the data support the conclusions?

Reviewer #1: Yes

Reviewer #2: Yes

2. Has the statistical analysis been performed appropriately and rigorously? 

Reviewer #1: Yes

Reviewer #2: Yes

3. Have the authors made all data underlying the findings in their manuscript fully available?

Reviewer #1: Yes

Reviewer #2: Yes

4. Is the manuscript presented in an intelligible fashion and written in standard English?

Reviewer #1: Yes

Reviewer #2: Yes

5. Review Comments to the Author

Reviewer #1: Feedback related to Review Questions 1 and 2:

1. The authors set out to evaluate implementation of a community-engaged approach to scale up Covid-19 testing in low-income, majority-Latino communities. They outlined a rigorous process for both community engagement and implementation evaluation, and provided a range of outcomes data to support the conclusion that such testing approaches rapidly increase access. Some amendments are suggested:

a. 12 patients who tested positive with antigen tests, but were asymptomatic and thus referred for confirmatory PCR—one was determined to have a false positive RDT and the other a false negative PCR. Please elaborate on the method used to make this distinction.

b. A concern with rapid antigen tests is their lower sensitivity; to evaluate this, confirmatory PCR for people with negative antigen tests but strong clinical suggestion of Covid-19 is sometimes recommended. Was this considered for any patients in the study and why/why not? Please elaborate in discussion.

c. Time was reported as a key factor in the decision to test/not, and the process took about twice as long in Canal (22min) vs Planada (10mins)—could you elaborate on the main contributors to this wide difference?

d. On a related note, Canal testing relied on hired medical workers whereas Planada used volunteers who’d been trained to do testing. Please comment on the role of non-medical workers in expanding community-based testing and related observations from implementation, if any.

e. While the study focused on Covid-19, its conclusions might potentially be relevant to other diseases for which expanded testing in under-served communities could be beneficial. Please address if relevant.

f. The ongoing maintenance of this approach is encouraging. Please address the extent to which different elements of the study, beyond testing itself (e.g. community mobilization, support for those testing positive) is included in the current efforts.

2. The authors described characteristics of the study population implementation outcomes in detail, providing ranges and confidence intervals where relevant. Limitations in the analysis, for example related to census data, were noted in the discussion. Multivariate logistic regression was used to identify factors associated with previous testing; as the OR and CI for correlation between higher age and never having been tested (1.02; 1.00—1.04) is very close to 1, it may be better to highlight only the factors with stronger associations.

Reviewer #2: This is a relatively well-written paper that describes a community-academic partnership to adapt San Francisco's Unidos en Salud "test and respond" model for community-based COVID-19 testing and post-test support in two US-census tracts: Canal (Marin county) and Planada (Merced county). The project utilized the RE-AIM framework to evaluate the project's progress/outcomes, being able to show the project's reach (albeit the evaluation sample sizes were not large). This is generally an interesting study that offers lessons relevant to jurisdictions that may be grappling with how best to reach hard to reach/culturally diverse, low-income populations. Below are some comments for the authors to consider:

(a) Mass testing/mass vaccination events have been used quite a lot during the pandemic but their cost-effectiveness is somewhat mixed and often not documented beyond the use of models. And in those studies that looked at comparative costs, they frequently place more weight on healthcare utilization and hospital costs than community and other operational costs - e.g., they often do not explicitly account for the costs of community engagement, recruitment, time, and the fact that in a health crisis there is stimulus funding supporting start-up activities (these are considerable) - in a lesser crisis or during an endemic situation, the stimulus funding support will not be available. Think the authors should expand on and discuss this a little bit more in the Discussion. It is an important limitation of the project and its conclusions.

(b) There is likely a lot of self-selection bias in those who attended the events (e.g., largely younger group [median ages were 32 and 38 for Canal and Planda, respectively] that are not at the highest risk of severe COVID disease and/or hospitalization/death - older age with comorbidities are clear risk factors for COVID-19). In addition, the positivity rates were only 3% and 1% in the two respective census tracts, and information on comorbidities and legal status (undocumented) were not consistently collected (albeit for good reasons, in the case of the latter due to the sensitive nature of the issue). These and other factors could use a little bit more exploration and discussion in the paper.

(c) More than half of the attendees were essential workers - this aspect of the project should be highlighted more and discussed further. It offers a good justification on why using an event-style approach to reaching Latino, low-income communities is not a bad idea.

6. PLOS authors have the option to publish the peer review history of their article (what does this mean?). If published, this will include your full peer review and any attached files.

Reviewer #1: **Yes: **Paula Ihozo Akugizibwe

Reviewer #2: No

---

## [Author Response · Author response to Decision Letter 0]

9 Aug 2022

Response to Reviewer and Editor Comments

Response: We have reviewed these templates and addressed these additional requirements in our revision.

 

Response to Reviewers' comments:

Reviewer's Responses to Questions

Comments to the Author

1. Is the manuscript technically sound, and do the data support the conclusions?

Reviewer #1: Yes

Reviewer #2: Yes

2. Has the statistical analysis been performed appropriately and rigorously?

Reviewer #1: Yes

Reviewer #2: Yes

3. Have the authors made all data underlying the findings in their manuscript fully available?

Reviewer #1: Yes

Reviewer #2: Yes

4. Is the manuscript presented in an intelligible fashion and written in standard English?

Reviewer #1: Yes

Reviewer #2: Yes

5. Review Comments to the Author

Reviewer #1: Feedback related to Review Questions 1 and 2:

1. The authors set out to evaluate implementation of a community-engaged approach to scale up Covid-19 testing in low-income, majority-Latino communities. They outlined a rigorous process for both community engagement and implementation evaluation, and provided a range of outcomes data to support the conclusion that such testing approaches rapidly increase access. 

 Response: We thank the reviewer for these comments.

Some amendments are suggested:

a. 12 patients who tested positive with antigen tests, but were asymptomatic and thus referred for confirmatory PCR—one was determined to have a false positive RDT and the other a false negative PCR. Please elaborate on the method used to make this distinction.

Response: We thank the reviewer for the suggested amendments, which further strengthen our manuscript. We now elaborate on how we made distinctions of false positive RDT and false negative PCR, with added detail in the Results section (see page 17, last paragraph, lines 264-269, as follows: 

“The participant considered to have a false-positive rapid test result was a child in a household with no known COVID-19 infections, who remained asymptomatic for 10-days post-testing, and had a negative confirmatory PCR result. The participant considered to have a false-negative PCR was an adult who tested positive on both repeat rapid antigen testing and repeat PCR confirmatory testing one day after the initial negative PCR result.”

b. A concern with rapid antigen tests is their lower sensitivity; to evaluate this, confirmatory PCR for people with negative antigen tests but strong clinical suggestion of Covid-19 is sometimes recommended. Was this considered for any patients in the study and why/why not? Please elaborate in discussion.

Response: We thank the Reviewer for this comment. Though we did not offer confirmatory testing for rapid antigen test negative participants in the mass testing campaigns, going forward in our “Maintenance” phase of community-based testing (post-campaign), we initiated confirmatory PCR testing for people with negative rapid antigen tests but strong clinical suggestion of COVID-19 (based on symptoms and/or close contact). We now include this in our Discussion section, when discussing limitations, as follows (see page 26, last paragraph, lines 437-444): 

“Lastly, during our mass testing campaigns, we did not offer PCR testing for persons with negative antigen tests who had symptoms concerning for COVID-19 or close contacts with persons who had COVID-19, and as such we may have missed some persons with early infection with antigen-negative, PCR-positive COVID-19. However, we have previously demonstrated that rapid antigen testing has a high sensitivity for identifying persons with low cycle thresholds on PCR, and in our maintenance phase of testing, we have since instituted PCR testing for persons with negative antigen tests who have symptoms concerning for COVID-19 or high-risk exposures.”

c. Time was reported as a key factor in the decision to test/not, and the process took about twice as long in Canal (22min) vs Planada (10mins)—could you elaborate on the main contributors to this wide difference?

Response: We suspect that the primary contributors to the differences observed in process time in the Canal compared to Planada were: a) the Canal was the first site of the two (with processes still being learned by the project team), and b) we observed a greater number of families and persons who self-registered in the Canal, whereas in Planada, a greater proportion of participants did not arrive with phones and as such received assistance from staff in registering (rather than self-registering) - a more rapid process (given event staff familiarity with the survey). 

We now elaborate on this difference in time from registration to completion of testing by site (Canal vs. Planada), in the Discussion section, as follows (see page 25, first paragraph, lines 402-411): 

“Third, our implementation measures of time from registration to testing suggest that such testing events can address the commonly reported barrier to testing of having “no time” to test, with rapid (10-22 minute) throughput. The relatively longer throughput (by 12-minutes) in the Canal vs Planada may have been attributable to the Canal being the first event site (with event processes still being learned by the project team) as well as relatively more families testing together in the Canal and self-registering (i.e., completing registration surveys on their own phones for multiple family members, thereby delaying time to completion of testing), though we are unable to verify this as an explanation for the differences in throughput time by site.”

d. On a related note, Canal testing relied on hired medical workers whereas Planada used volunteers who’d been trained to do testing. Please comment on the role of non-medical workers in expanding community-based testing and related observations from implementation, if any.

Response: In our Discussion, we now comment on the role of non-medical workers in expanding community-based testing and note that this type of task-shifting has since been implemented at the Canal site – further supporting low-barrier, community-based testing adoption and implementation. Please see page 25, lines 415-418: “In addition, the use of local, non-medical workers to conduct testing in Planada was a key adaptation that has since been expanded to the Canal site and may further support rapid testing expansion in times of increased testing and longer-term maintenance.”

e. While the study focused on Covid-19, its conclusions might potentially be relevant to other diseases for which expanded testing in under-served communities could be beneficial. Please address if relevant.

Response: We thank the Reviewer for this comment, and strongly agree. Indeed, our research group has further explored the addition of other diseases to COVID-19 testing campaigns in San Francisco: we now discuss this further in our Discussion (please see pages 25-26, lines 421-424), and include a new citation to integration of diabetes screening into community-based COVID-19 testing (reference 23, Kerkhoff, et al) as follows:

“Such approaches also hold the promise of expansion to address other high-priority prevention and screening services, via multi-disease service offering, as the Unidos en Salud project has demonstrated in San Francisco, and merit further investigation.”

f. The ongoing maintenance of this approach is encouraging. Please address the extent to which different elements of the study, beyond testing itself (e.g. community mobilization, support for those testing positive) is included in the current efforts.

Response: In the Results section, we now address the extent to which support for those who test positive has been maintained (indeed, this support is ongoing), as well as additional elements of the study, beyond testing, have been maintained. Please see the Results section, page 20, lines 321-331, as follows (with red font indicating updated text): 

“Following testing events and post-event sampling in both communities, low-barrier (i.e., walk-up with no appointments or identification required), community-based rapid COVID-19 testing sites offering rapid antigen testing with PCR confirmation, as well as post-test support services, were maintained by local CBOs (Canal Alliance and United Way-Merced) weekly throughout 2021 and remain ongoing at the time of publication. In the Canal, the community-based testing site has also served as a local COVID-19 vaccination site, with walk-up vaccination or vaccine appointments, on COVID-19 testing days. More recently, facilitated linkage of persons with COVID-19 to outpatient therapy at local clinics has also been implemented from the community-based testing sites in both the Canal and Planada. The LCC continues to meet monthly as well, at the time of publication, supporting community mobilization efforts and communication as the COVID-19 pandemic and public health response continues to evolve.”

2. The authors described characteristics of the study population implementation outcomes in detail, providing ranges and confidence intervals where relevant. Limitations in the analysis, for example related to census data, were noted in the discussion. Multivariate logistic regression was used to identify factors associated with previous testing; as the OR and CI for correlation between higher age and never having been tested (1.02; 1.00—1.04) is very close to 1, it may be better to highlight only the factors with stronger associations.

Response: We thank the reviewer for this comment. Although the association may not seem as strong as other factors, this association is “per year” of age. Furthermore, we provide the confidence interval for readers, and age is an important variable within the multivariate model: on this basis, we do think it important to provide this association in the Results. Of note, we do not emphasize this finding in the Discussion (see page 24, first paragraph, lines 380-392), relative to other factors associated with no prior testing. 

Reviewer #2

Reviewer #2: This is a relatively well-written paper that describes a community-academic partnership to adapt San Francisco's Unidos en Salud "test and respond" model for community-based COVID-19 testing and post-test support in two US-census tracts: Canal (Marin county) and Planada (Merced county). The project utilized the RE-AIM framework to evaluate the project's progress/outcomes, being able to show the project's reach (albeit the evaluation sample sizes were not large). This is generally an interesting study that offers lessons relevant to jurisdictions that may be grappling with how best to reach hard to reach/culturally diverse, low-income populations. Below are some comments for the authors to consider:

(a) Mass testing/mass vaccination events have been used quite a lot during the pandemic but their cost-effectiveness is somewhat mixed and often not documented beyond the use of models. And in those studies that looked at comparative costs, they frequently place more weight on healthcare utilization and hospital costs than community and other operational costs - e.g., they often do not explicitly account for the costs of community engagement, recruitment, time, and the fact that in a health crisis there is stimulus funding supporting start-up activities (these are considerable) - in a lesser crisis or during an endemic situation, the stimulus funding support will not be available. Think the authors should expand on and discuss this a little bit more in the Discussion. It is an important limitation of the project and its conclusions.

Response: We thank the Reviewer for this comment. In our Discussion section, when discussing limitations, we now further highlight this point when discussing limitations of our project. Please see page 26, lines 425-429, as follows: “The costs associated with such testing events include costs related to community mobilization efforts, personnel time and laboratory testing costs, as well as government stimulus funds to assist persons testing positive for COVID-19 that may not continue as the pandemic progresses.”

(b) There is likely a lot of self-selection bias in those who attended the events (e.g., largely younger group [median ages were 32 and 38 for Canal and Planda, respectively] that are not at the highest risk of severe COVID disease and/or hospitalization/death - older age with comorbidities are clear risk factors for COVID-19). In addition, the positivity rates were only 3% and 1% in the two respective census tracts, and information on comorbidities and legal status (undocumented) were not consistently collected (albeit for good reasons, in the case of the latter due to the sensitive nature of the issue). These and other factors could use a little bit more exploration and discussion in the paper.

Response: We agree with the Reviewer that persons who attended the events may have differed from the underlying population, and we believe one strength of our study is that we have addressed this issue in several ways. First, we surveyed a random sample of testing event non-attendees to gain insights into potential selection bias. Second, we provide comparative demographics to the underlying census tract of each community (see Table 2). For example, despite the relatively young age of testing event attendees (compared to those at highest risk of death from COVID-19), the event attendees were older than the median age in each census demographic, and indeed, the events reached persons at relatively higher risk of poor outcomes (including lower-income and predominantly Latino community members). Furthermore, we include discussion of limitations of using census tract denominators, particularly as it relates to undocumented persons (see Discussion section, page 26, lines 431-434). 

Lastly, although the positive rates were relatively 3% and 1%, this was likely a result of when the mass testing events occurred (a time of relatively low incidence in the US in April and May 2021) and is anticipated with community-based (rather than clinic-based) screening, where a relatively higher proportion of asymptomatic persons may be reached. To provide context for this, we include the positivity rates at that time from each county’s Department of Public Health in our Results. We now also include the potential for under-ascertainment of persons with antigen-negative but PCR-positive COVID-19, to further explore possible ascertainment bias in positivity rates, in our limitations paragraph (please see page 26, lines 437-444), as follows: 

“Lastly, during our mass testing campaigns, we did not offer PCR testing for persons with negative antigen tests who had symptoms concerning for COVID-19 or close contacts with persons who had COVID-19, and as such we may have missed some persons with early infection with antigen-negative, PCR-positive COVID-19. However, we have previously demonstrated that rapid antigen testing has a high sensitivity for identifying persons with low cycle thresholds on PCR, and in our maintenance phase of testing, we have since instituted PCR testing for persons with negative antigen tests who have symptoms concerning for COVID-19 or high-risk exposures.”

(c) More than half of the attendees were essential workers - this aspect of the project should be highlighted more and discussed further. It offers a good justification on why using an event-style approach to reaching Latino, low-income communities is not a bad idea.

Response: We thank the Reviewer for this suggestion. We now highlight the high proportion of frontline (essential) workers reached by our testing events in the Discussion, with references to the disproportionate impact of the pandemic on this group of workers. Please see page 23, lines 369-371, as follows: “We ensured testing event days included weekends and evening hours to optimize access for frontline workers. Likely because of these efforts, most persons who tested (63%) at the events were essential, frontline workers – a group disproportionately affected by the COVID-19 pandemic.” In addition, please see page 25, lines 401-405, as follows: “Third, our implementation measures of time from registration to testing suggest that such testing events can address the commonly reported barrier to testing of having “no time” to test, with rapid (10-22 minute) throughput. Indeed, the high proportion of low-income residents and frontline workers suggests that such events may have addressed this barrier for this at-risk population.”

---

## [Decision Letter · Decision Letter 1]

4 Oct 2022

Reproducibility and Implementation of a Rapid, Community-Based COVID-19 “Test and Respond” Model in Low-Income, Majority-Latino Communities in Northern California

PONE-D-22-13602R1

Dear Dr. Chamie,

We’re pleased to inform you that your manuscript has been judged scientifically suitable for publication and will be formally accepted for publication once it meets all outstanding technical requirements.

Kind regards,

Benedikt Ley, PhD

Academic Editor

PLOS ONE

Additional Editor Comments (optional):

Reviewers' comments:

Reviewer's Responses to Questions

**Comments to the Author**

1. If the authors have adequately addressed your comments raised in a previous round of review and you feel that this manuscript is now acceptable for publication, you may indicate that here to bypass the “Comments to the Author” section, enter your conflict of interest statement in the “Confidential to Editor” section, and submit your "Accept" recommendation.

Reviewer #1: All comments have been addressed

Reviewer #2: All comments have been addressed

2. Is the manuscript technically sound, and do the data support the conclusions?

Reviewer #1: Yes

Reviewer #2: Yes

3. Has the statistical analysis been performed appropriately and rigorously? 

Reviewer #1: Yes

Reviewer #2: Yes

4. Have the authors made all data underlying the findings in their manuscript fully available?

Reviewer #1: Yes

Reviewer #2: Yes

5. Is the manuscript presented in an intelligible fashion and written in standard English?

Reviewer #1: Yes

Reviewer #2: Yes

6. Review Comments to the Author

Reviewer #1: Question 1: All my comments have been satisfactorily addressed and I have no further queries. I noted with interest Reviewer #2's first comment on cost effectiveness analysis, which is indeed a common limitation of such projects in many settings. Among populations experiencing significant access barriers e.g. migrant documentation, it can be particularly challenging to analyse the cost/benefit of the counterfactual and account for the likelihood that some individuals might simply not access testing in the absence of such an intervention, with corresponding effects on transmission etc. It's a complex task that could be good for the authors to explore further during the maintenance phase of the campaigns-- but with the revisions made to this paper and acknowledgement of limitations, the conclusions they have drawn for this particular intervention still hold.

Question 4: Data will be made available upon acceptance.

Reviewer #2: The authors have adequately addressed this reviewer's comments from the previous round of reviews. There are no concerns about dual publication, research ethics, or publication ethics. Thanks.

7. PLOS authors have the option to publish the peer review history of their article (what does this mean?). If published, this will include your full peer review and any attached files.

Reviewer #1: **Yes: **Paula Ihozo Akugizibwe

Reviewer #2: No

---

## [Editor Report · Acceptance letter]

18 Oct 2022

PONE-D-22-13602R1 

Reproducibility and implementation of a rapid, community-based COVID-19 “test and respond” model in low-income, majority-Latino communities in Northern California 

Dear Dr. Chamie:

I'm pleased to inform you that your manuscript has been deemed suitable for publication in PLOS ONE. Congratulations! Your manuscript is now with our production department. 

Kind regards, 

on behalf of

Dr Benedikt Ley 

Academic Editor

PLOS ONE